# Heidegger as Levinas's Guide to Judaism beyond Philosophy

**Elad Lapidot**

Faculty of Theology, University of Bern, 3012 Bern, Switzerland; elad.lapidot@theol.unibe.ch

**Abstract:** This essay reflects on the way that Emmanuel Levinas stages the difference between Judaism and Philosophy, namely how he approaches Jewish thought as a concrete other of philosophy. The claim is that this mise en scène underlies Levinas's oeuvre not only as a discourse about the Other, but as a real scene of an actual encounter with otherness, namely the encounter of philosophy with the epistemic otherness of Judaism. It is in the turn to Jewish thought beyond Philosophy that the essay identifies Heidegger's strongest influence on Levinas. The essay's reflection is performed through a reading of Levinas's first major philosophical work of 1961, Totality and Infinity. The encounter between Philosophy and Judaism is explored in this context both as an epistemic and as a political event.

**Keywords:** Heidegger; Levinas; totality and infinity; philosophy; Judaism; politics





The name Heidegger has come to designate a singular phenomenon in the history of thought. The combination of powerful philosophical insight with powerful political shortsightedness (to say the least) has been producing what we can call tormented reception, namely generations of thinkers feeling uneasy about how deeply they were indebted to Heidegger. So uneasy, since their most ingenious attempts of taking distance from Heidegger were still inspired by his thought, and so brought them again and ever closer to him. Especially concerned were Heidegger's direct students, and above all the Jewish ones, such as Emmanuel Levinas.[1] Indeed, the paradoxes arising from tormented reception of Heidegger may be suggested as a key for reading many of Levinas's major contributions. In this essay I wish to focus on one of these paradoxes of reception, perhaps the greatest one, since it concerns the specifically Jewish aspect of Levinas's work.

Scholars have indicated two fundamental ways in which Levinas was inspired by Heidegger, paradoxically, precisely where he sought to depart from Heidegger, namely in thinking the Other beyond Being, ethics beyond ontology. One strand of scholarship has pointed out how central elements of Levinas's endeavor to rethink philosophy (responsibility, original ethics, constitutive intersubjectivity, foundational language etc.) are already present in Heidegger's own revision of philosophy.[2] A second group of scholars accentuated the paradox of Levinas's anti-Heideggerian Heideggerianism by indicating how Heidegger's thought (may have) inspired Levinas's "Other" not only as a thematic or structural configuration *within* (reformed) philosophy, but as a configuration that is itself an Other *of* philosophy, namely arising from an intellectual tradition than is different than Greek philosophy: Judaism. Before Choachani, it was Heidegger who opened Levinas to Jewish thought as an alternative to the anti-Jewish Heidegger. Heidegger was Levinas's first rav. Daniel Herskowitz recently argued that Levinas used Heidegger's method of *Dasein* analytic to counter Heidegger's "pagan" phenomenology with a phenomenology that draws on Jewish existence.[3] Michael Fagenblat went even further to further to argue that not only the phenomenological method, but Levinas's very turn to Jewish thought as a source for philosophy was inspired by Heidegger's reconfiguration of philosophy as based on historical hermeneutics (a feature which in his turn Heidegger unavowedly drew, as Marlène Zarader claimed, from "Hebraic" thought).[4]

My own endeavor in this essay takes an additional step in arguing a Heideggerian inspiration for Levinas's thought of the Other not only as an alternative (i.e., not ontology-

based, but ethics-based) philosophy, but more specifically as arising from Jewish thought as alternative to philosophy. Herskowitz, Fagenblat and Zarader understand Levinas's turn to the intellectual otherness of Jewish tradition as an attempt to use it for a reconfiguration of philosophy, which would ultimately bridge the difference between philosophy and Jewish thought. What Levinas provides, Fagenblat writes, is neither "Judaism *and* philosophy", nor "*between* Athens *and* Jerusalem", but rather "philosophy of Judaism without *and* or *between*".[5] As my following analysis will show, I agree that Levinas provides *also* this. However, I'm interested in how Levinas, before uniting Judaism and philosophy, preliminarily stages the difference between Judaism *and* Philosophy (I make a point of writing Philosophy with a capital letter, to emphasis the historical specificity of this intellectual tradition), namely how he approaches Jewish thought as a concrete *other* of philosophical thought. My claim is that this *mise en scène* underlies Levinas's oeuvre not only as a discourse about the Other, but as a real scene of an actual encounter with otherness, namely the encounter of Philosophy with the epistemic otherness of Judaism. It is in the turn to Jewish thought *beyond* Philosophy that I identify Heidegger's strongest influence on Levinas.

Jacques Derrida already indicated the ambiguous relations of Levinas's epistemo-political critique of Western Philosophy to Heidegger's.[6] Levinas on the one hand portrays Heidegger as the epitome of Philosophy's totalitarian ontology, on the other hand reproduces central aspects of Heidegger's critique against the tradition of Western Philosophy as promoting knowledge in the form of theory, and so suppressing individual action. It was precisely Heidegger's existential wake-up call that in the 1920s attracted to him young students such as Levinas and Arendt, or Hans Jonas, who similarly to Levinas was drawn away from Husserl's rigorous science of *Bewusstsein* to Heidegger's existential analytics of *Dasein*. Levinas himself was aware of his debt to Heidegger when, for instance, in his first major work of philosophy, *Totality and Infinity* of 1961, he acknowledged that ontology does not have to serve totality, but "[i]n its comprehension of being . . . it is concerned with critique."[7] Indeed, Heidegger did not deploy the notion of Being to confirm the universal order of beings as established by the philosophical tradition, but on the contrary in order to question it.

Levinas went on to qualify his acknowledgement of the critical potential of ontology by indicting that "[i]ts critical intention then leads it beyond theory and ontology". In other words, Philosophy has the potential of self-critique that would take it beyond itself—towards its epistemic other. This inter-epistemic or *trans*-epistemic intention was also central for Heidegger's project, insofar as his basic point in calling our understanding of Being into question was to argue that *different* understandings—and so different ontologies, different epistemologies—are possible, of which Western philosophy only represents a specific one. The basic vector of Heidegger's thought was to go beyond—to "destroy"—this tradition towards a radically different way of thinking and knowing, a radically different epistemology, which would no longer be epistemology (focused on scientific knowledge) nor ontology (focused on beings), and in later Heidegger it would not even belong any more to Philosophy, but to alternative historical locations, such as the pre-Socratic Greeks or Hölderlin's postmodern Germans.

Heidegger's trans-epistemic thought will look for non-Philosophy not only before or after the West, but also parallel to it, for instance in East Asia.[8] In his earliest lectures, however, years before *Being and Time*, he developed his critique of Western philosophy and laid the foundations for his trans-epistemology from a location of non-Philosophy situated within the West, and speaking Greek, but with an accent, a Judean one, namely the epistles of the Greek Jew Paul, also known, for Christians, as the Bible. It is in this same direction, from West to East, which will also proceed the trans-epistemic quest of Heidegger's student Hans Jonas, observing beyond Platonic theory a different, foreign, existential kind of knowledge, called in Greek not *episteme*, but *gnosis*. It is also in this direction that points Levinas's *Totality and Infinity* when it confronts Philosophy, a Western tradition of epistemic totality and political totalitarianism, with the alternative tradition of "the Prophets".

The present essay attempts a reading of *Totality and Infinity* as a trans-epistemic project of strong Heideggerian inspiration. Levinas's first major philosophical book of 1961 does not stand at the end, but at the beginning of his venture beyond Philosophy, which will culminate in his long engagement in the non-philosophical corpus of the Talmud. The Talmudic Readings are the primary site in Levinas's work where a concrete configuration of epistemic difference is explicitly and methodically staged. There is a deep structural kinship between Heidegger's attempts to overcome the archive of Western philosophy through reading Hölderlin and Levinas's move from philosophy to the Talmudic page.[9] Not the least, for both thinkers, the non- or anti-philosophical archive serves as a primary site for their engagement not only with epistemology, but with politics—both conceptually and with respect to current events. This essay follows the epistemo-political question in pointing out Heideggerian motifs also in Levinas's early trans-epistemic endeavor, in *Totality and Infinity*, which, so I claim, seeks to transcend Philosophy within the corpus of Philosophy.[10]

The following analysis is divided in two parts: the *first part* articulates the basic elements for reading *Totality and Infinity* as an inter-epistemic project, namely as staging an encounter between two different conceptions, systems and traditions of knowledge, between two epistemes; the *second part* proposes a critical reading of the phenomenological narrative offered by Levinas in *Totality and Infinity* as the scene of an inter-epistemic event. Arguing that in fact *no* such event takes place in Levinas's main narrative, my analysis locates the actual event of inter-epistemic difference, both split and encounter, at the very last sections of Levinas's book, as a *political* event, with a surprisingly Heideggerian plot. The essay concludes by offering a critique of this plot, which points at the basic epistemo-political paradox of the difference discourse that generates a discourse of identity.

## 1. Between Philosophers and Prophets

That inter-epistemic encounter is at the heart of *Totality and Infinity* is not evident. The epistemic difference is not completely hidden in the book, but it is also not conspicuous. In Levinas's philosophical discourse in general, in contrast to his Jewish writings, the drama of inter-epistemic encounter remains for the most part insinuated, alluded to in conceptual argumentation or phenomenological description, abstracted and allegorized. In the language of Philosophy, Levinas speaks of the encounter between Philosophy and non-Philosophy often in riddles, always somewhat ambiguous, somehow enigmatic, in code.

Take for instance the title, *Totality and Infinity. An Essay on Exteriority*. This title places side by side two nouns, similar to two names of two separate traditions, Totality and Infinity, interrelated by a mere "and", by a minimal relation of juxtaposition, a relation of "exteriority", as the subtitle suggests. Yet these are not names, but concepts, philosophical terms, categories within one epistemic system, Philosophy, which is too present and obvious to be named. The juxtaposition of "Totality and the Infinite" can be easily read not as disjunction but as conjunction, as bringing together what belongs together, what is basically the same, similar to *Being and Time*. The "and" suggests no difference, but rather the coherence of a panoramic view.

And yet, from the outset, Levinas describes the book as seeking to expose something often perceived as the inner tension within one and the same system, in the essence of culture or the nature of man, that is the tension between the True and the Good, between knowledge and morality, theory and praxis, contemplation and action, between Kant's first and second critiques, as a "split" between two separate traditions, as an inter-epistemic difference:

"To tell the truth, ever since eschatology has opposed peace to war the evidence of war has been maintained in an essentially hypocritical civilization, that is, attached both to the True and to the Good, henceforth antagonistic. It is perhaps time to see in hypocrisy not only a base contingent defect of man, but the profound split [*déchirement*] of a world attached at the same time to both the philosophers and the prophets." (TI, 9, 24)

What appears to be one, what *is* already one world, ours, would in fact be the indecision between two different, separate and antagonistic poles. This bi-polar tension is built not just on difference, which may also (such as in Hegel) constitute the inner structure of a systematic whole, but on a split, disruption and disturbance of wholeness, the rupture of totality. Our world is torn between two incommensurable civilizations, which are not just two concepts or terms conjoined in one discourse, not just two philosophical, Latin categories, *totalitas* and *infinitas*, nor even two modes of consciousness, but two separate figures of humanity, marked by two antagonistic paradigms of epistemic agency, "the philosophers and the prophets".[11] These two paradigms are both designated by Greek categories, trace accordingly an inner tension in a world built on Greek, ours, the West. At the same time however they invoke one of the most common inter-epistemic distinctions operative in—and generative of—Western discourse, namely between Greek and non-Greek cultures of knowledge, and more specifically between Athens and Jerusalem, Plato and Moses, Greek and Jewish.

The tension between "philosophers and prophets" preserves the ambiguity of the epistemic difference around which the book's argument revolves; it is a difference both within one world and between two, both a conceptual tension between Totality and Infinity and at the same time a split between two civilizations, two historical, textual and political projects, two separate traditions of knowledge and praxis, two epistemes.

The ambiguity between split and difference, between intra- to inter- is foundational for Levinas's staging of the inter-epistemic encounter in his philosophical writings. The philosophical discourse is only able to refer to its epistemic other in code, to acknowledge it not by proper name but always in concept, always already meaningful, already understood, domesticated, de-othered. The reference is often made not by direct designation, but by incidental association. In *Totality and Infinity*, the idea of the Infinite, easily translated—through Levinas's explicit reference to Descartes—by "God", is occasionally associated, for instance, with the "monotheistic faith" (TI 75, 77), as well as with ideas such as "creation *ex nihilo*" and "sabbatical existence" (TI 107, 104). Philosophy's epistemic other is also associated with "religion" (TI 58, 107, 331), but Levinas explicitly declines to identify it with theology, for ambivalent reasons, whether because it is too philosophical (TI 326) or too foreign to philosophy (TI 106, 332).

In contrast, it is in no ambiguous terms that Levinas rejects the designation of non-philosophical thought, which his book promotes, as "oriental". He fends off this epithet—"an alleged Oriental thought" (TI 105, 330)—as an accusation, an insult. This is without doubt a reaction to an entire modern discourse of Secularism, Orientalism and Semitism, a mix of anti-Semitic slur and philo-Semitic condescendence, which has been fulfilling precisely the role of devaluing the epistemic tradition that Levinas asserts, the role of de-epistemizing it.[12] However, what Levinas seems to reject even more fundamentally in "Oriental" is not the adjective, but the name, not because what it means, for instance non-Greek, but because it doesn't have any epistemic meaning, it is not a concept, but a name, it does not mean but simply points—to the East. In contrast to its Western counterpart, which in Levinas's texts commonly serves to determine philosophy ("Western philosophy"), Oriental otherness is non-epistemic. West would relate to East not as one episteme to another, but as episteme to no-episteme.[13]

Accordingly, both times Levinas rejects "oriental" as improper designation for the philosophers' epistemic other, he immediately assures us of the "dignity" of this alternative tradition of thought by associating it with a well-known motif in the constitutive text of the Philosophy tradition, Plato's Idea of the Good. "That there could be a more than being or an above being", Levinas writes of a notion that he believes exceeds Philosophy, "is expressed in the idea of creation which, in God, exceeds a being eternally satisfied with itself. However, this notion of being above being does not come from theology. If it has played no role in the Western philosophy issued from Aristotle, the Platonic idea of the Good ensures it the dignity of a philosophical thought, and it therefore should not be traced back to any oriental wisdom." (TI 241, 218). Next to Plato's Idea of the Good, Levinas

defends the philosophical dignity of non-philosophy by identifying it also in a second central figure of Philosophy, a modern one, that is Descartes' Idea of the Infinite, which gives to non-Philosophy its philosophical codename, "Infinity".

What I examine as an "inter-epistemic" encounter, between Philosophy and non-Philosophy, the philosophers and the prophets, takes place in Levinas's first major philosophical opus for the most part in the form of an inner-philosophical drama. "Totality and the Infinite" reads initially as the tension between two currents within the philosophical tradition: a mainstream, "from Parmenides to Spinoza and Hegel" (TI 105, 102) or "from Plato to Heidegger" (TI 327, 294), and an undercurrent, a subversive tradition, which only surfaces at specific moments, for instance in some Platonic or Cartesian ideas. The translation of the split between two traditions of knowledge, an inter-epistemic split, into an intra-epistemic tension within Philosophy, has the effect of converting philosophical concepts into codenames for the divide between Philosophy and non-Philosophy, between the Greek and the Jewish: ontology and metaphysics, Totality and Infinity.

### What Is the Difference about?

Under the title distinction between Totality and Infinity, Levinas offers multiple characterizations of these two epistemic paradigms. I highlight two central features. First, all epistemic characterizations concern the question of difference itself, such that the two different worlds of knowledge represent two traditions of dealing with difference, they differ on difference, they are "others" to each other because each has another approach to otherness. Second, the epistemic difference between these two conceptions and performances of difference is articulated by the internal relation, within each one of the epistemes, between knowledge and practice. To be sure, the question concerning the ethical and political aspects of knowledge is central for the entire Levinasian project. *Totality and Infinity* is motivated by the tension between wisdom and morality from its first line: "Everyone will readily agree that it is of the highest importance to know whether we are not duped by morality." (TI 5, 21)

The tradition of wisdom is, in this constellation, Philosophy. Levinas's basic characterization of the philosophical episteme is "totality". As a constellation of difference, totality signifies a multiplicity of different elements that nonetheless constitute a whole. Totality is difference as unity of the different. Philosophy, the tradition of totality, is "philosophy of unity", promoting "the ancient privilege of unity that is affirmed from Parmenides to Spinoza and Hegel" (TI 105). Accordingly, the basic operation of philosophy is generating totality by overcoming difference in unity.

Levinas indicates two main epistemological paradigms that have been serving Philosophy in generating totality: theory and ontology.

Theory, from the Greek *theorein*, meaning "looking, viewing, observing", signifies knowledge that is based on seeing, on vision. Both theory and vision, as well as light, have an ambivalent epistemic role in *Totality and Infinity*.[14] On the one hand, theory is knowledge from distance, which not only respects difference, but could constitute the essence of the epistemology of difference, the opposite of totality—and of Philosophy. On the other hand, theory also has a second meaning, which is exactly opposite to the first: "a way of approaching the known being such that its alterity with regard to the knowing being vanishes." (TI 32, 42). The ambivalence of theory is the ambivalence of light. Light is a medium of nothingness, which enables knowledge from distance. On the other hand, due to the same quality, light is also absolute medium, overcoming all distance and so connecting all discrete points into one total vision, one see-all, what Levinas calls "the panoramic" (TI 328, 294). It is this totalizing effect that Levinas identifies as the epistemic core of Philosophy, which "from Plato to Heidegger" has been thinking and performing knowledge as an operation of "bringing to light" (TI 327, 294). By a Platonic metaphor, the epistemically strongest light for Levinas is not sunlight, sensual light, but intellectual light, reason or logos. The logos that Philosophy has been deploying to generate total visibility is the "logos of being" (TI 32, 34), ontology.

Levinas has here in mind Heidegger's notion of "Being" (*Sein*), which is what we necessarily always already understand in order to enter into whatever relation with anything, namely as something that exists, that *is*. Being is the light we must already see in order to see everything else. In the light of Being all things, as different and diverse as they may be, nonetheless *are*, and so are already accessible, knowable, visible as beings, which in Heidegger's *Being and Time* are mostly totalized into "the being" (*das Seiende*). As noted above, Heidegger is analyzed in *Totality and Infinity* as an accomplished and especially articulated version of Western philosophy, which "has for the most part been ontology" (TI 33, 43), and as such, episteme of totality.

Levinas's central critique of this episteme is that by abolishing all difference, it leaves no epistemic room for individual beings. Totality makes the individual insignificant. The obfuscation of individuality concerns not only the object of knowledge, the known being, the Other. The main thrust of Levinas's critique of epistemic totality is that it more fundamentally abolishes the individuality of the knower—of the Self. The ultimate epistemic consequence of totality, as overcoming difference, is not the subjugation of all otherness, things and persons, to my own selfish and solipsistic knowledge, but the absorption of both the others and myself, of all selves, into impersonal knowledge, the disappearance of all individuality in Being, which "destroys the identity of the Same". (TI 5, 21)

Levinas's project is animated by the *ethical* implication of this epistemic destruction of identity in totality. Levinas describes the basic practical implication of totalizing thought as generating "movement", namely of bringing individual identities into synch with the total vision: "a casting into movement of beings hitherto anchored in their identity, a mobilization of absolutes, by an objective order from which there is no escape." (TI 5, 21) This loss of identity is how Levinas defines the essence of violence, of evil.

For Levinas, the evil of total knowledge lies in abolishing individuality and thereby destroying the basic condition for human action as arising from individual, moral agents, who are responsible for their actions. Totality negates morality by abolishing individual responsibility. *Totality and Infinity* begins by describing how "lucidity" (knowledge guided by light) knows a reality that is without morality, a world that is ontologically violent, at a "state of war". "The state of war suspends morality; it divests the eternal institutions and obligations of their eternity and rescinds ad interim the unconditional imperatives." (TI 5, 21) Eternal obligations are based on unconditional imperatives, which for Levinas require absolute individual responsibilities—eternity or infinity, which is correlative to individuality. For total knowledge, in contrast, there is no meaning for individual decisions. The only judgment that Philosophy would know is not moral judgment, but History.

The paradigmatic form of praxis that arises from Philosophy's epistemology of totality, the praxis that performs the impossibility of action, the praxis of non-morality, of war, is for Levinas "politics". Politics would be the performance of totality to the exclusion of individuality. Whereas for Hannah Arendt 20th century total politics was epitomized in the image of the Movement,[15] Levinas imagines totality as "the tyranny of the State" (TI 37, 46). As a State, the human world of action is without individuals, it is anonymous, impersonal. When Levinas describes ontology as a "philosophy of power" (TI 37, 46) or a "philosophy of injustice" (TI 38, 46), when he speaks about "ontological imperialism" (TI 35, 44), the basic immorality of these constellations lies not in selfish disrespect of others, but in the loss of the individual self, of the responsible moral agent. Even though the term is not mentioned in *Totality and Infinity*, the most adequate designation for politics as Levinas imagines it in this book, as the practical correlate of Philosophy's epistemological totality, is indeed Arendt's concept of "totalitarianism".[16]

In contrast to Philosophy, *prophetic epistemology* stands for the *rupture* of totality. If Totality –the epistemological principle of Philosophy—is the constellation of difference as overcome in unity, prophetic knowledge maintains difference in non-unity. It pertains to the essential paradox of Levinas's project that he presents this constellation of difference, which marks a rupture with the epistemic totality of ontology, in ontological terms, as the

"ultimate structure of being" (TI 104, 102). Non-totalized difference means "being" that is "produced as multiple and as split in Same and Other" (TI 269, 301).

Formally, Levinas characterizes the difference he has in mind as a relation where "[th]e terms remain absolute despite the relation in which they find themselves." (TI 197, 180) Levinas designates this relation of absolute difference with the category *separation*. Levinas identifies the idea of separation in one of the basic categories attributed to the prophetic tradition, that is *creation*, "in which the kinship of beings among themselves is affirmed, but at the same time their radical heterogeneity also, their reciprocal exteriority coming from nothingness." (TI 326, 293)

Based on the paradigm of separation, prophetic epistemology cannot be ontology, because it cannot be guided by any uniting logos, any "light", that is it cannot be based on knowledge as vision: "otherwise the Same and the Other would be reunited under one gaze, and the absolute distance that separates them filled in." (TI 24–25, 36) Accordingly, knowledge of difference cannot lie outside of difference, but must pertain to the relation of difference: knowledge of separation must perform separation. This kind of knowledge does not consist in positioning entities relative to one another in indifferent space, it is not a conjunction of one "and" another. Rather, the relation of separation constitutes a movement or vector that goes from one separated being towards another, as the "orientation of being 'from oneself' towards 'the Other'" (TI 237), as "being for the other" (TI 340).

Accordingly, the epistemology of difference is focused on the separated entity, which constitutes, by virtue of its separateness from others, the non-other, namely the Same, the entity whose being consists in remaining identical to itself: the Self. Conceptually, and this is a fundamental paradox of any philosophy of difference, rigorous commitment to difference requires equally rigorous commitment to identity. For Levinas, being identical to itself, being self, is what constitutes subjectivity, such that his anti-totalitarian project, which consists in affirming separation, is also described as "defense of subjectivity" (TI 11).[17] The concrete existence of the self, according to Levinas, is the individual self, which therefore constitutes the absolute perspective of knowledge, and is therefore designated as the individual knowledge subject, the singular first person "Me".[18]

Levinas describes the Me as "interiority". The relation of separation signifies interiority that is open to the outside. *Totality and Infinity*, a defense of subjectivity, is therefore "An Essay on Exteriority". Exteriority is otherness as "known" from the self's inner point of view. In prophetic epistemology knowledge constitutes the relation of an inner self "with a surplus that is always exterior to totality" (IT 7), namely knowledge as *revelation*. The exterior, the other, is that towards which the self's knowledge is oriented. Levinas calls it "radical" exteriority, which means that it can never be interior, since it constitutes the absolute exteriority in relation to which—and in separation from—interiority may exist as such. Levinas designates this exteriority by the preposition "beyond", *au-delà* or "transcendence", meta-. Most prominently, however, in his first book Levinas designates the otherness of knowledge, which essentially remains beyond it, by the Cartesian term "Infinity", the book's titular counter concept of Totality and so the philosophical codename for Prophecy.

Levinas's central point is that prophetic knowledge is in essence not a relation of vision, not theory. Separation is rather generated by knowledge that constitutes an actual "being for the other", which does not *see* the other, but is seized by the other, and *goes* towards the other, outside of itself. This characterization immediately calls to mind Heidegger's conception of human existence as ex-sitence, namely as based on ek-statis, being "outside-of-itself" (BT 329, 377). The knowledge of otherness, relation to infinity, is characterized by Levinas as "desire" or "attitude": "an attitude already specified as love or hatred, obedience or command, learning or teaching, etc." (TI 126, 121) The episteme of difference is not based on ontology, but on axiology. In this episteme, to know, to see, is already to act: "ethics is optics" (TI 8).

Levinas's central contribution to the epistemology of difference is to have indicated that difference-based knowledge cannot exist in the indifferent space of theory, of true

and false, but lives in ethics, namely in the dimension of good and bad. Not only this kind of knowledge, the basis of epistemic diversity, does not exclude morality, such as the totalizing paradigm of Philosophy, non-totalizing knowledge *is* in essence moral and so the very nature of this episteme transcends the "opposition between theory and practice." (TI 15, 29).[19]

Beyond ethics, Levinas identifies another essential feature of any epistemology of difference, *language*. "Absolute difference", he writes in *Totality and Infinity*, "inconceivable in terms of formal logic, is established only by language. Language accomplishes a relation between terms that breaks up the unity of a genus. The terms, the interlocutors, absolve themselves from the relation or remain absolute within relationship. Language is perhaps to be defined as the very power to break the continuity of being or of history." (TI 212, 195) Levinas's work points at a deep affinity between ethics and language as central features of difference-based knowledge, which would be constitute the episteme tradition of the Prophets.

## 2. The Philonic Encounter

So far, I have presented the inter-epistemic difference that underlies *Totality and Infinity*, which translates, in the language of Philosophy, the split between philosophers and prophets, between Philosophy and non-Philosophy. The book, however, is not a mere indication of this difference, but an intervention in it. It features Levinas's earlier attempt to perform, within Philosophy, an encounter with non-Philosophy, a trans-epistemic event.

### 2.1. Deduction

We now understand that approaching the difference between Philosophy and non-Philosophy from the point of view of Philosophy is no contradiction but required by the epistemology of difference as outlined above; the Other must be approached from the Same, and not observed from a presumed neutral point of view.

How is this approach accomplished? The epistemology of difference requires not a theoretical but an ethical approach, desire. However, the designation that Levinas gives for his movement from the philosophical to the prophetic episteme, from totality to infinity, belongs to a highly theoretical discourse: "deduction". Levinas calls it "the phenomenological method" (TI 14, 28), referring to Husserl.

This method, Levinas explains, departs from vision-based, objectifying thought to "reveal" it as "implanted in . . . a forgotten experience from which it lives": "The break-up of the formal structure of thought . . . into events which this structure dissimulates, but which sustain it and restore its concrete significance, constitutes a *deduction*—necessary and yet non-analytical.'" (TI 14, 28). De-duction leads from theoretical knowledge *back* to something else—horizon, experience, event, situation–, from which theory "lives". Just as Husserl showed our perception of objects to arise from experiences of our own conscience, Levinas wants to show how *totalizing* knowledge lives from non-totality: "we can trace back the experience of totality to a situation where totality breaks up, a situation that conditions totality itself." (TI 9–10, 24)

What must be noted at the outset is the ambivalence of this exercise: by tracing back totality to non-totality, namely by deducing non-totality from totality, it establishes a *necessary* connection between totality and non-totality and thus overcomes their difference, to reunite them in a new, even more comprehensive totality. More specifically, it presents non-totality not as the opposite of totality, but as its foundation. What Levinas undertakes as resistance to totality, i.e., showing that it is conditioned by difference, at the same time features difference as the basis *of* totality.[20]

From the formal structure of Levinas's intervention follows its concrete operation. Showing totality to be based on non-totality means integrating the two into a comprehensive total discourse, a total narrative. Levinas identified the paradigm of such a total logos as "the logos of being", and his demonstration—his "deduction"—in *Totality and Infinity*, inasmuch as it mounts an opposition to ontology, is itself accordingly deeply

ontological, looks for the "ultimate structure of being" (TI 104, 102). Prophecy is revealed as the condition of Philosophy, but this revelation itself is made on Philosophy's terms. Levinas's deduction may therefore be describes as Jewish Greek in the sense that it speaks the language of a Jewish Philosopher, it constitutes a Philonic operation.[21]

Levinas's explicit reference and model is not Philo, but Husserl. Its total logos is not only a logos of being, ontology, but also a logos of phenomena, phenomenology, namely an observation, description and depiction of reality. This reality, contemplated as incarnating the ontological plot, is paradigmatically the singular individual subject, a "me", which defines a primary domain of the real, "interiority" or "experience". The conceptual movement (the "deduction" of non-totality from totality) is accordingly described as a process that takes place in the subject, some development in its condition. The phenomenological demonstration proceeds as a narrative of this event within the inner experience of the self, which emerges as the overarching totality of non-totality with totality.

This model is prominent in classics of modern philosophy that are directly referenced by *Totality and Infinity*, from Heidegger's *Being and Time*, through Husserl's *Ideen II*, Hegel's *Phenomenology* and Descartes' *Meditations*. Levinas's text, in its trans-epistemic performance, renders tangible the affinity of these philosophical narratives to prophetic narratives, such as the biblical myth. In the narratives of modern philosophy, the individual subject embodies the totalizing form of the narrative, which integrates the Other in the Same, God in the world, Beyng in beings. Accordingly, and this is visible also in Levinas's narrative, the individual drama represents a broader, collective or political event, namely functions as parable for history. Phenomenology epitomizes historiography.[22]

What I argue is that Levinas's story, more than similar phenomenological narratives, is built on its own struggle to be a story, namely not only the illustration of logical relations, which arise from formal necessity, but an occurrence of otherness, within totality a breakup of totality, a real encounter. Levinas often designates this happening by the term "event" or "situation", which implies the lack of necessity, contingency, incommensurable with any theory or objective calculation. The difficulty is however obvious: the phenomenological description of the non-necessary event is supposed to perform the *necessary* deduction of non-totality from totality. If we take this difficulty as a hermeneutic key for reading Levinas's narrative, then, beyond its uncontestable poetic beauty and insightfulness and against Levinas's own structural articulation, his narrative can be shown to fall in two parts, which articulate a—structural, inexplicit—self-reflection of Levinas's inter-epistemic project on itself.

*The first part* is the main part of the book, which traces back "the experience of totality to a situation where totality breaks up," through a phenomenology of subjectivity. Since the experience of totality is narrated as what constitutes subjectivity, the key moment of this narrative is the subject's encounter with the Other as the event *par excellence*. However, since this event is *deduced* from totality, rather than disrupting, it arises as the basis of totality. The main part of the book thus in fact demonstrates rather the unity of Totality "and" Infinity: how Greek philosophy is based on Prophetic ethics, how our world is not split between them, but arises from their unity. The *second part* of Levinas's narrative, which takes place in the last sections of the book, is consequently dedicated to showing how the problem of totality arises from the Greek-Jewish *unity* of ontology and ethics. It is here that the real story transpires not as the encounter with otherness, but as the history of this encounter's epistemo-political abuse.

### 2.2. Plato and Moses Meet Descartes

The first part of Levinas's narrative begins in the "experience of totality". This notion, and the description to which it gives rise, are built on equivocation. On the one hand, this is an experience of totality; on the other hand, it is an *experienced* totality, namely that is already predicated on individual subjectivity, and therefore on difference. Accordingly, Levinas portrayal of totality is a phenomenological description of the individual self, the "me", whose basic principle is to exist as an individual, separate being. *Totality lives from*

*separation.* The main thrust of Levinas's narrative is to insist—against any narrative that makes the individual self dependent on a higher order—on the independent being of the individual subject, whose existence is defined by contentment, enjoyment and happiness. Against Heidegger and original sin theology, human being is not "fallen" in worldly life. Levinas's self is so to speak a Greek one, who is basically "at home" in the world, who exists, so the title of this section, as "Interiority and Economy".

And yet, inasmuch as the self is independent of unworldliness, it depends on the world. Existing in the world as an individual means subsisting *by means of* the relation to worldly otherness. This is the essence of enjoyment: breathing, eating, reproduction. The self maintains itself through its relation to worldly others, which are accordingly *relative* others, only existing as a moment in the self-generation of the individual subject. Separate individual existence consists in a dialectic relation between self and world, which is by definition a relation of non-separation. Inasmuch as Levinas describes a separate being, this description is at the same time designed to portray the "experience of totality", which means the effacement of the self, namely self-effacement by the very force of self-generation through the world.

This dialectical logic—separation as totalization—underlies Levinas's rich narrative, which portrays subjective experience as an evolution between two fundamental conditions. In the first, primal stage, the individual has more immediate, sensual relations to the world, which is experienced not as objects or things, but as "elements". The individual is "bathing in" the elements: enjoying himself (male gender explicit) through them, living from them, but at the same time in constant insecurity and fear of losing himself in them, of becoming them, not dying, but being absorbed into the anonymous "there is".

In the second, higher evolutionary stage of the individual's self-identification through the world, the subject establishes relations, beyond the elements, with another subject, another person, who is however encountered not as absolute other, but as a part of the subject's own world. Levinas's identifies this inner-worldly otherness as "the feminine", who is the familiar other, a familiar "you", a *tu*.[23] The relation to the feminine opens up within the elemental world a dimension of familiarity, where the individual finds his interior space out in the world: Home. The home world is no longer the chaotic world of elements, but regulated existence, Economy, the "first civilization" (TI 163). Economic civilization "adjourns" elemental immediacy, so that the individual no longer just "bathes in elements" but acquires hold of things. By the same dialectics of enjoyment, however, the things are only relative others, which exist "for me" and not "in themselves", as mere phenomena and not as actual beings. The economic world is still a form of interiority, in which self-identification generates an "experience of totality", which signifies the effacement of the self in its own world. One is reminded of *Dasein*'s being-in-the-world, in Heidegger's *Being and Time*, where the human subject "initially and most often" loses himself in the world he dwells.

The main point of Levinas's demonstration is that totality is always "experience of totality", namely a subjective experience, a state of the separated self. Totality is a performance of difference. It follows—this is the *deduction*—that totality requires, as a necessary pre-condition, separation, "a situation where totality breaks up, a situation that conditions totality itself." This situation is the self's encounter with the absolute Other.

My point is that, even as Levinas describes this passage from the "experience of totality" to non-totality, to the relation with the absolute Other, as a "situation", "a new event" (TI 185), "new energy" (TI 183) or even as a moment of "grace" (TI 161), namely as transcending all necessity and logic; and even as this contingency is what enables the narrative, descriptive, phenomenological quality of Levinas's demonstration, it nonetheless remains a conceptual argument, a deduction, which operates with the force of logical necessity. There is a structural correlation between the experience of totality, interiority and the relation to the Infinite, to exteriority. Totality and the breakup of totality articulate one and the same constellation. This is the hermeneutic key that I propose for reading the

centerpiece of *Totality and Infinity*, the "situation where totality breaks up", the encounter with the Other.

The encounter with the Other is portrayed by Levinas as a redeeming event of knowledge. This encounter shows parallels to other mythical revelations of redeeming knowledge, such as the "call" that in Gnostic myths or in Heidegger's *Being and Time* wakes up the self from worldly self-oblivion. *Totality and Infinity* explicitly invokes as a reference, within the philosophical archive, the Cartesian ego's overcoming of doubt by the idea of the infinite. Another explicit acknowledgment reminds here also of the second stage in Rosenzweig's narrative in *The Star of Redemption*, the specifically *biblical* revelation, which is indeed Levinas's primary reference, namely the prophetic myth.

The center of Levinas's demonstration is in fact the appearance of biblical prophecy, as an explicit quote, in the world of totality, the emergence of Jewish transcendence in Greek immanence. What Levinas describes as the "situation where totality breaks up" features prophecy as an episteme of difference, characterized by the two elements that I indicated above: *ethical* relation of self to other, which takes place in *language*. It is revealing how Levinas's phenomenology modulates these basic elements such that prophecy appears as enabling the totalizing epistemology of Western philosophy. A central motif in this constellation is how the relation to the other, ethics, operates as the constitutive event in the existence of the separate, independent self.

The encounter with the Other is an event of language. For Levinas, the paradigmatic phenomenon of language, which is the element of prophecy, is the act of *spoken* language, the living word, *parole*. Language, as the original relation to the Other, is a relation between speakers, interlocutors. Language originally appears as voice. The Other appears as "voice coming from the other bank" (TI 186). The voice is the Other's presence to me. For Levinas, this vocal presence is the original experience of presence, of being, namely of something that exists not for me, but for itself, *an sich*, objectively. In *Totality and Infinity*, otherness, paradoxically, means presence, being. The relation to the other is the relation to something present. Accordingly, in Levinas's depiction of this relation, the auditory experience of voice is famously translated into the encounter with a paradigmatically visual object, with the *face*. Language is phenomenologically a relation of vision.

Here lies the significance of Levinas's notion that, in the epistemology of difference, ethics is optics. The encounter with the Other is the original event of objective knowledge. The Other is the primal object and so the primal source of knowledge, namely, as Levinas writes, *le Maître*, "the Master" or "the Teacher". Knowledge is essentially "teaching". Even as Levinas invokes in this context Descartes (God teaches me the idea of infinity) against Plato (the teacher only helps me to recall what I already know) (TI 85), his analysis of teaching and the teacher clearly echoes "torah" and "rabbi" as two basic Jewish epistemological categories.

Yet, the basic teaching of the Other, the basic knowledge that he dispenses, does not lie in what he says, but in the voice, which means nothing but itself, namely absolute being. The revelation of absolute being in the inner experience of the self, which is an experience of totality, of permanent effacement of the difference between self and other, means an encounter with *resistance* to self-identification through others. The encounter with the other's absolute being means an encounter with something that may not be dialectically made part of the self. As Levinas puts it, it is an encounter with the possibility of "total negation of a being", with something that is not only relatively, but absolutely different than me, which has an absolutely independent existence, and so which I—through my constant appropriation of the world—may completely negate, annihilate. The concrete phenomenon of absolute negation of being is killing. When Levinas therefore writes that "[t]he Other is the only being that I may want to kill" (TI 216), this arises analytically from the notion that the Other is the only absolute being. Being is being exposed to killing, which means that the Other, as absolute being, is encountered as absolute vulnerability. Levinas expresses this by invoking a prophetic trope, describing the Other not only as teacher, but also as "the foreigner, the widow and the orphan" (TI 237).

Consequently, the self's experience of the other, as absolute being, is an experience of resistance to the self's own being, to his self-identification through others. The absolute other, as an absolute object, is encountered as objection, as opposition, as a "no". "No" is the original word spoken by the face, the original prophecy, first *torah*. In *Totality and Infinity* the original speech act that constitutes the encounter with the Other appears as a quote from the paradigmatic instance of biblical prophecy, the sole direct speech of the Divine to the entire community, the Ten Commandments. The core of the Ten Commandments, God's primal word—this is the "no" that Levinas quotes—would be, to follow *Totality and Infinity*, the 6th commandment: לֹא תרצח (*lo tirtzach*; Exodus 20, 12), "You shall not commit murder" (TI 217).

Positing Commandment Six as First Prophecy is not obvious.[24] In Levinas's terms, "killing" arises as the possibility of absolute negation, namely of the only being that absolutely *is*, the other. This possibility, the essence of my encounter with the other, is experienced as a resistance to my own being, a "no", which marks an end to my power, to my possibilities. The Other is experienced as impossibility. As Levinas makes clear, this limitation of my power is not imposed by a stronger power, against whom I am too weak, but in contrast by absolute weakness. The other is experienced as something that is essentially beyond my power, that I cannot access with power, that I may only access through self-limitation. Self-limitation generates the experience of "ethical impossibility" (TI 185). This experience constitutes moral conscience, the knowledge of good and bad, which Levinas refers to Plato's idea of the Good. In this ethical optics, certain entities—speaking others—may only be encountered—known –*as* self-limitation of my power, namely as objects with respect to whom I *should* not act. Their very being, their objectivity, their resistance to me, their "no", is a commandment: "you shall not". This limitation of my action does not mean this act is not in my power, but that it is bad, it is violence. It is only in the optics of ethics that we see certain acts as crimes: killing is murder. The basic experience of the other as absolute being is the encounter with my possibility of killing him, which is experienced as a negative commandment, "You shall not commit murder".

The phenomenological analysis of the Sixth Commandment as articulating our fundamental experience of being, as fundamental ontology, is one of Levinas's most powerful and famous interventions. In the narrative of *Totality and Infinity*, it marks the moment in which, within the subject's interior "experience of totality", totality breaks up by the presence of the absolute other. Since the experience of totality is the existence of the self, as self-identifying in (relative, worldly) others, the breakup of totality means the limitation of the self, such that he experiences his power as violence, which essentially implies self-negation, a restrained, moral attitude of "no violence"—ethics.

Nonetheless, and this is my point, according to the logic of Levinas's deduction, this moment of totality's break up is also the *condition* of totality. It is crucial to note the necessary function of ethical self-restraint in the structure of separation, which means the being of the self. Nietzsche pointed out the self-empowering force of ascetic morality. In Levinas's plot too, the encounter with the Other, as an event of self-limitation, is exactly the emergence of self-conscience, of the explicit experience of being an individual, the experience of "Me". A fundamental insight of Levinas is that the encounter with the infinite Other is the encounter with an infinite invocation—assignation, summoning, accusation or "election"—of the Self, as infinite correlation to the Other, as infinite responsibility.[25]

This is how the situation where totality breaks up, the ethical encounter with the Other, generates the individual self, whose immanent experience is the condition of totality. However, as revelation goes, the emergence of the separate self as the condition of totality does not merely lead to a repetition of the "experience of totality", but to a higher level of experience, to self-conscience.

The first part of Levinas's story thus concludes with the developed form of his epistemology of difference. Of Jewish origins, this epistemology is nonetheless visibly Greek. It is founded on the constitutive relation to the absolute other, which, as we saw, is built on the paradigm of objective knowledge, and has the structure of vision. Accordingly,

even as epistemic difference exists in language, just as original language, living speech, means presence of face, encounter with objective being, developed language constitutes the language of objectivity, language as logos. Prophetic revelation would be the foundation of theory, "divine veracity that supports Cartesian rationalism" (TI 224). Rationalism is featured in *Totality and Infinity* as the very performance of ethics, since, Levinas explains, generalization means generosity, the "offering of the world to the other person" (TI 189).

And so, Levinas's epistemology of difference, which seemed to stand as a subversive, prophetic, Jewish alternative to Greek epistemology of totality, reveals itself rather as a fusion of Moses and Plato, a Philonic vision, whose modern embodiment, in *Totality and Infinity*, is the Judeo-Greco-French Descartes. "What is Europe?", Levinas will write a quarter of a century later, "It is the Bible and the Greeks".[26]

### 2.3. The State against God's People

This vision features in Levinas's narrative as the conclusion of a drama, the completion of the psychological development of the individual subject, or of the historical formation of Western civilization. Yet, as already noted, notwithstanding the dramatic language, Levinas's phenomenological portrait, by its deductive methodology, in fact outlines a static constellation of necessary correlations, the constellation of separation. Within the structure of separation, the emergent relation to absolute otherness, in the face of the other person, generates the absolute point of departure for this relation, namely the separate individual self, who exists as interiority. In other words, up to this point, which is the great part of *Totality and Infinity*, Levinas's narrative does not really feature a story, but a point of departure for one. What this story needs to tell is the birth of totality, not as the "experience" of individual subjectivity, but as a historical episteme, as a tradition of knowledge, Western Philosophy, which stands not only upon but *in opposition* to Prophecy. The last part of Levinas's narrative is called to tell the history of how the episteme of totality arises *from* the episteme of difference.

I therefore claim that the actual drama of *Totality and Infinity* takes place in its last sections, and goes, as the last section is titled, "Beyond the Face", namely beyond the momentary "situation" of the individual encounter between self and other. Its concern is indicated in the title of the immediately preceding subsection, "The Ethical Relation and Time". This may be read as a direct conversation with Heidegger's *Being and Time*. Levinas here acknowledges that the phenomenological constellation described in the first part of his narrative, which culminated in the Other's revelation in inner experience as the ethical event that generates the separated self, Levinas acknowledges that this constellation, the ethical relation, in order to exist, must persist, namely be in time. The second part of Levinas's narrative thus deals with the episteme of difference not as a structure of the individual conscience, but as a culture or world of knowledge, as a historical episteme, a civilization.

One point should be carefully noted, concerning the trans-epistemic happening. My initial reading of *Totality and Infinity* suggested that this book stages a confrontation between two distinct traditions of knowledge, two conflicting epistemes, Totality and Infinity, Philosophers vs. Prophets, Greek vs. Jewish. My analysis of Levinas's narrative, however, produced a more complex picture. The point of departure for history—time—in Levinas's narrative is not a conflict between two opposite epistemes, Greek and Jewish, but a Jewish-Greek(-French) episteme, a fusion of Moses and Plato, Philosophy founded on Prophecy, moral rationalism. This means that the actual inter-epistemic drama in *Totality and Infinity* does not play between Greek and Jewish, but between two different performances of their composition, between two different configurations of the West.

The possibility of multiple simultaneous performances of the same constellation requires a certain contingency, a moment of indeterminacy within the conceptual structure. The possibility of multiplicity, where something such as event or history, something such as time, takes place, this possibility, the possible par excellence, marks the real location of otherness in the narrative. The emergence of otherness is a constitutive moment in all

narratives, since it established the very possibility of story, the mythical foundation of myth. This event, as the emergence of contingency, of freedom, is marked by a moment when things do not work as they should, where something happens that should not, when things go wrong. History begins with evil. Biblical mythology is based on a story of sin and fall. Fall, *Verfall*, is also a foundational moment in Heidegger's narrative, set in motion by *Dasein*'s fall into improper existence, *Uneigentlichkeit*. *Totality and Infinity* uses a different Heideggerian category, more epistemic one: "forgetfulness".

The possibility of *forgetting* the other arises from the nature of otherness. The separate individual, constituted in correlation to the other, *can* forget the other. Levinas identifies this possible oblivion of otherness as the possibility of forgetting God, "atheism" (TI 188, pp. 172–73, 197, 181), which attests to the very power of creation, creating a creature so independent it is capable of forgetting its creator.

In Levinas's analysis, by forgetting the other, the self closes on itself, oblivious to ethics, generating "the possibility of injustice and radical egoism" (TI 188, 173). Here lies the origin of evil and of history. However, the historical evil that *Totality and Infinity* is concerned with is the rise of totality, namely the *disappearance* of the individual ego. This is the epistemo-political pathology that Levinas diagnoses in Western Philosophy "from Plato to Heidegger"—total logos and total state. One of the important insights developed in *Totality and Infinity* is that totality, the forgetting of individuality, arises from self-identification. *Totality arises from individualism*. This was visible in the "experience of totality" generated by self-identification through others. The same dynamics now repeats itself in the new dimension opened by the relation to otherness, namely in the realm of reason. Forgetting the ethical foundation of reason, the uninhibited self perverts reason to an instrument of self-identification—to totalizing ontology. This perversion constitutes the being of the separate self in time, "beyond the face", as history. Philosophy, episteme of totality, would be the historical episteme of Jewish-Greek knowledge that forgot its Jewish origins, its foundation on ethical transcendence and reestablished immanence.

This episteme of totality would be the first, problematic historical configuration of difference epistemology, its perverted version. The pathology appears in both constitutive dimensions of difference epistemology, namely in language and in ethics. We saw that "language" is Levinas's most fundamental characterization of the medium in which the episteme of difference exists. Separation is a relation of language. Accordingly, the perversion of this relation is a perversion of language. It is language that enables evil, forgetfulness, history, totality. *Totality and Infinity* identifies the perversion of language in *writing*. Whereas the Other is absolutely present to me in living speech, in the voice, written language would be the diminished form of language. The pathology of spoken language, the origin of evil, would be the sign. "The sign", Levinas writes, "is a mute language, an impeded language" (TI 199, 182). Built on absence, the sign is the site of negativity.[27]

The perversion of language, writing, at the same time perverts ethics. *Totality and Infinity* identifies the fallen form of morality in politics. Language as writing, as "work", Levinas notes, constitutes "the tyranny of the State" (TI 191, 176).[28] The State, the *polis*, what Levinas calls here "politics", would be the perverted form of justice. The problem is totality. The State is the materialized manifestation of impersonal logos, which overcomes difference in the total system, and so "reduces all ethics to politics" (TI 239, 216), generating "a tyranny of the universal and of the impersonal." (TI 271, 242) The historical figure that comes to mind here is Rome, the Judeo-Greek Empire and closer to Levinas, all the phenomena analyzed by Hannah Arendt as 20th century Totalitarianism.

The State is the perverted relation to the Other in time, the disfiguration of the Jewish-Greek, prophetic-philosophical episteme. It seeks to fulfill ethics through historical reason, which abolishes all individuality. Against this pathology, Levinas proposes an alternative, a more authentic West, committed to difference. This figure functions as the telos, the destination of Levinas's demonstration. It emerges as the dramatic denouement, the rectification of evil, the return of the fallen, a—happy—end of history. In Heidegger's narrative this eschaton appears as proper existence, *Eigentlichkeit*, in Rosenzweig's drama as

Redemption. In Levinas's plot, this ultimate figure embodies the non-perverted, authentic performance of the Jewish-Greek episteme of difference. In opposition to Christian Rome, the enactment of Otherness in Hellenic means, we presume here a more Jewish enactment, a dissenting prophetic agency within the ontologized West. In contrast to the temporal performance of the ethical relation to infinity in the improper form of the State, as politics, *Totality and Infinity* identifies authentic ethical existence in time as *religion*.

Against the impersonal tyranny of politics, "the religious order" is "where the recognition of the individual concerns him in his singularity." (TI 271, 242) The essence of "religious conscience", Levinas writes, is the acknowledgment of a moral judgment outside history, a "judgment of God" (TI 273). Against time as totality, history, the time of states, religion requires a configuration of time as non-total, temporality that is predicated on infinity, an "infinite and discontinuous time" (TI 336). Infinite time means "infinite being", which is the temporal being of the Infinite. The infinite being of the Infinite is the infinite being of the individual self's *relation* to infinity, the relation that constitutes individuality. Infinite time implies the infinite being of the separate self, the infinite individual. The infinite individual is individual beyond finitude, beyond the finite, mortal, singular self, beyond "Me". The individual beyond the singular is the *plural* individual self. Accordingly, religion is the dimension in which the relation to the other, ethics, is enacted in time, beyond the singular individual, not as a State, but as a plural Subject, a plural self, a "We". The authentic performance of the prophetic episteme of difference would be a We, a *people*. Indeed, against the common reading of Levinas's "religion" in *Totality and Infinity* as based on individual ethics, I suggest that this is the authentic dimension of collective existence, of society and also of politics.[29]

It is instructive to note how Levinas's epistemic configuration of the prophetic collective performance, to which the last section of *Totality and Infinity* is dedicated, challenges the structure of separation, which this configuration is nonetheless called to perform. This redeeming episode takes the narrative back to its earliest stage, before the situation where totality broke up, before the encounter with the voice, before the face, before the absolute Other, before revelation. We are taken back to interiority. More precisely, we are taken back to the encounter with the *relative* absolute other, with the Feminine, who is the other that remains interior to the self's experience, the other who is no master, no *vous*, but a familiar *tu*. It is in the familiar relation of the (essentially masculine) subject to the woman, before absolute separation, which infinite being is generated, where infinity comes into being.

Let us look how this realm of being is configured with respect to the two basic features of Levinas's epistemology of difference, language and ethics. The feminine face does not speak. It signifies by its "feminine beauty", which signifies the lack of signifying, and therewith the "disfigurement" of the face (TI 294; 263). What the feminine face expresses is "its renunciation of expression and speech" (TI 295, 263). The feminine other is encountered in silence. Since otherness exists as present in the voice, the silent relation to the feminine brings the subject into contact with no presence, with no absolute being, which means that it is no longer—or not yet—a relation of separation.

The ethical essence of this relation, as an attitude of the self toward the other, is not performed in moral self-restraint, in "no", but in voluptuosity, in sexuality, in eros. In Levinas's analysis, eros supersedes separation, unites the self and the other, man and woman, to generate the child, who is "at the same time other and myself" (TI 298). In fecundity, relating to the other, transcendence, means becoming the other, "transubstantiation" (TI 298). This is how the singular individual becomes a plural individual, becomes family. Family is the existence of individuality beyond the finite individual, it is the infinite individual being, the "ultimate structure" of being, "produced as multiple and as split in Same and Other" (TI 301). As a counter-vision to the temporal performance of ethics as politics, namely as State, which is a universal totality, with no individuality, Levinas posits the performance of ethics as the plural individual subject, the "we" of the family. "The family does not only result from a rational arrangement of animality; it does not simply

mark a step toward the anonymous universality of the State. It identifies itself outside of the State, even if the State reserves a framework for it." (TI 342, 306)

The family is the ultimate figure of *Totality and Infinity*, as the paradigm of authentic ethical existence in time, not as a total State but as a plural individual self-identifying Subject, as "We". In other words, the family is the paradigm of collective ethical existence that is not based on the category of the *polis*, not "political", but instead based on the notion of the people as a collective individual subject, what in modern categories is often called *nation*. Levinas does not use this category here[30], and as I noted the common reading situates Levinas family in the sphere of individual or private ethics, whose political corollary, "fraternity", is simply universal bond of all humankind. Nevertheless, it is my contention that the Family functions in *Totality and Infinity* as the epitome of the people as the infinite collective *individual* being of the infinite, namely of God's people. God's people is the multitude that becomes individualized, unified in specific collective self-identity, by being collectively subjected to commandments, by election for infinite responsibility for the Infinite, chosen for the Good, for God. Levinas does not say this explicitly, but once again his text can be decoded to indicate, against a Greek state-based, a Jewish people-based social thought and against a Greek performance of the Jewish-Greek episteme of difference in the historical-political figure of Rome, a Jewish performance in the figure of Israel, whose historicity is not properly speaking historical, and whose politics is not properly political. This reading has the advantage of connecting *Totality and Infinity* to the entire discourse of "Israel" in Levinas's Jewish writings, with the political complexity of messianic "universalist particularism".[31]

It is here—in meta-politics—that the surprising affinity to Heidegger appears.[32] Levinas's positing of the family as a counter-figure to the state, a collective subject against a total object, indeed calls to mind the famous §74 of *Being and Time*, where Heidegger portrays *Dasein*'s authentic existence as "destiny", namely—in contrast to the inauthentic, objectified and de-individualized "they" (*man*)–, "the event [*Geschehen*] of the community, of the people [*Volk*]" (BT 384, 436). This passage has become the main piece of evidence brought up by critics of Heidegger to prove his early attachment to nationalism.[33] Heidegger too does not use the term "nation" in this context, and §74 offers different, more nuanced readings of the nature of the community Heidegger is speaking of, which do not necessarily lead to and *völkisch* ideology.

And yet, thinking with Levinas himself, whose entire project explicitly seeks to turn away from Heidegger, as an alleged representative of the entire Western philosophy, based on the paradigm of unity, we should ask whether positing the "family" as a corrective to the state does not raise fundamental difficulties. Indeed, Hannah Arendt, whom I already paralleled to Levinas as a contemporary anti-totalitarian thinker, certainly did not consider the genealogical conception of the collective subject, the nation-family, as resistance to totalitarianism, but on the contrary as being, through the category of "race", the main ideological vehicle of modern totalitarian movements. She showed how it was precisely the combination of divine election ("God's people") with statelessness that made the Jewish people into a source of inspiration for race-based imperialism.[34] Arendt identified this inspiration as perversion, but her answer to this imperial perversion was not a better politics of chosen people. On the contrary, her response to Western totalitarian imperialism was a return to the Greek politics of *polis*, which is no infinite being, but a territorially—and metaphysically—limited state.[35]

It is noteworthy that *Totality and Infinity* provides a last, very brief indication towards a horizon in which the notion of the plural subject as infinite being regains the structure of totality. The collective self remains—this is the whole point—an individual self, which, by the very logic of separation, exists in self-identification, namely in generating—a collective— "experience of totality". "Truth", Levinas writes in the last lines of his narrative, "demands both an infinite time and a time that it may seal—a completed time. The completion of time is not death, but messianic time where the perpetual is converted into eternal. Messianic triumph is pure triumph. It is secured against the revenge of evil whose return the infinite

time does not prohibit. Is this eternity a new structure of time, or an extreme vigilance of the messianic consciousness?" (TI 318, 284–85).

"Truth" is the epistemic core of religion, as the people's collective self-conscience. It demands more than the sense of surpassing finite individual destinies, more than "infinite time". It also demands "a completed time", namely a notion of ultimate purpose, a destination, an *eschaton*, an end of time. In this vision infinity becomes eternity. Redemption is complete and final, total. "Pure triumph" leaves no place for "evil", which is however, as I indicated in Levinas's own narrative, precisely the Other in logos. Levinas leaves open— "the problem exceeds the bounds of this book"—whether this total vision of eternity is "a new structure of time, or an extreme vigilance of the messianic consciousness", namely—to read these words with Rosenzweig in mind[36]—a new Christian Israel or an eternal Jewish people.

### 3. Conclusions

The basic logic of Levinas's narrative in *Totality and Infinity* is that totality and individuality do not contradict each other, rather totality is precisely the experience and very being of individual, separate self-identity. This is why separation can be deducted from the experience of totality. Accordingly, the totality of the state too, inasmuch as it erases singular individuality, can be and must be at the same time imply a radical *performance* of individual identity, an amplified form of subjectivity. Therefore, it is as *complementary* to the State that I suggest seeing the subjective figure of the people that Levinas presents in the concluding movement of his narrative as opposite and corrective to the State, as redemption from politics. The State is the state *of* the plural individuality of a "We".

As Arendt argued, the notion of God's peoples, of the religious, eschatological, messianic collective, did not contradict state totalitarianism, but provided the foundation and paradigm for the totalitarian, imperial subjectivity. In the history of the West, the Judeo-Christian people of Israel have in fact for the most part not been the enemies of Rome, but its citizens. This means that *Totality and Infinity*, by tracing back totality to the prophetic episteme of difference, to the chosen family, rather than overcoming the problem of Western totalitarianism, identified its origins. This realization will become the guiding *self-critical* insight of Levinas's all later attempts to understand and to intervene in the inter-epistemic tension between Philosophers and Prophets, both in his philosophical works and in his Talmudic readings. A critical aspect of this self-critique, of this *Kehre*, will be the actual and explicit "break up" of the totality of philosophical discourse and the emergence of the Talmud as its concrete, textual epistemic Other.

**Funding:** This research received no external funding.

**Institutional Review Board Statement:** Not applicable.

**Informed Consent Statement:** Not applicable.

**Conflicts of Interest:** The author declares no conflict of interest.

### Notes

[1] For an illuminating recent discussion of Heidegger's Jewish reception, where Levinas indeed plays a paradigmatic role, see (Herskowitz 2021).

[2] See various contributions in (Drabinski and Nelson 2014), such as Philip J. Maloney's (on the "secularization of transcendence"), Emilia Angelova (on temporality), Robert Bernasconi (useless sacrifice), François Raffoul (responsibility), Peter Gordon (nativism), Krzystof Ziarek (human dignity, critique of power); see also (Wolfson 2014, chp. 3), for a thorough and detailed demonstration of the affinities between Levinas and Heidegger on a long series of central asepcts of their philosophies.

[3] Herskowitz 2021, p. 256.

[4] (Fagenblat 2010), for instance p. xiii. See also (Zarader 2006).

[5] Fagenblat 2010, p. 14.

[6] (Derrida 1967, pp. 117–228).

7   (Levinas 1961, p. 33; Lingis 1969, p. 43). All citations from this work below will specify the acronym of the English title ("TI") followed by the page numbers in the French and then in the English edition.

8   See for instance, (Heidegger 1985, pp. 79–146).

9   Levinas explicitly invoked this analogy when he suggested that "Biblical verses . . . have as much right as Hölderlin and Trakl to be cited", see (Levinas and Levinas 2006, p. 66).

10  A longer study currently in preparation expands this investigation to the broader context of Levinas's oeuvre, including his second major philosophy book, *Otherwise than Being*, and the various collections of Jewish writings and Talmudic readings.

11  (Klemm 1989, pp. 403–26), identified not two, but three different "voices" in *Totality and Infinity*: "Levinas writes under three distinct yet often overlapping signatures. I call them the philosophical, the religious, and the prophetic signatures or voices." (p. 407)

12  On the discourse of "Semitism" and its de-epistemizing effect, see (Lapidot 2020).

13  The question of Levinas's Orientalism and more broadly problematic relationship to non-Western cultures, has been discussed by different authors in the last decade, see for instance (Moten 2018), and (Drabinski 2011).

14  This was one of Derrida's central observations in "Violence et métaphysique ", see for instance pp. 125–37.

15  (Arendt 1979, p. 427).

16  For a thoughtful reflection on the affinity and complementarity between Arendt's political thought and Levinas's ethics, see (Topolki 2015).

17  I thus concur with Leora Batnitzky that Levinas's "central argument in *Totality and Infinity* is for a separable, independent subject", see (Batnitzky 2006, p. 30).

18  This notion clearly echoes Heidegger's fundamental characterization of *Dasein*'s existence as "in each case mine, *je meines*", see (Heidegger 2001; Macquarrie and Robinson 1962, p. 67). Following references to this work will be made in the text, to "BT", with the page number of the German edition, following by the page number of the English translation.

19  As already noted, within Philosophy, Levinas indicates Plato's idea of the Good beyond Being as a subversive emergence of ethical knowledge within a basically ontological episteme. Beyond Philosophy, he evokes the notions of revelation and "teaching", which can be translated to *torah*. I suggest that within Greek knowledge-discourse, Levinas's notion of ethics-based knowledge, founded on the acknowledgment of radical otherness, is akin to what Hans Jonas identified with the category gnosis. For Jonas, gnosis means knowledge that is essentially—in Levinas's sense—ethics. Gnosis is how the forbidden knowledge in the constitutive ethico-epistemic myth of the prophetic discourse, "knowledge of good and evil" (Genesis 2, 17), was called in its Greek translation, γνωστὸν καλοῦ καὶ πονηροῦ. See Hans Jonas, *The Gnostic Religion*, S. xviii. On Levinas and Jonas, see (Vogel 2001, pp. 121–48).

20  This logic was the heart of Husserl's exercise, which sought to show how perception of objects is founded in intention towards objects. Levinas calls this the "literal" meaning of Husserl's method, which he, Levinas spiritualizes so as to lead perception back even behind objectal, theoretical intention, to a more fundamental, non-objectal experience of difference. However, by doing so Levinas does not just extend or deepens Husserl's deduction, but twists its logic. To base totalizing knowledge on totalizing intention is straightforward, to base totality on non-totality less so. See Derrida's critique in "Violence et métaphysique", p. 128.

21  Fagenblat in his book presents a compelling demonstration of how to read Levinas as a 20th century pendant of another paradigmatic Jewish philosopher, namely Maimonides, see Fagenblat 2010, passim. The paradigm of Philo, which I do not discuss in detail here, has in this context the advantage of constituting a preliminary intellectual connection between Athens and Jerusalem, before the more complex intellectual history of Maimonides, who was already facing at least three rich traditions of Greek Jewishness, namely the Christian, the Islamic and also the rabbinic.

22  Elliot Wolfson offered a precise analysis of the dynamics by the force of which attempts, such as Levinas's, of thinking transcendance beyond all imagination end up in new, powerful forms of figuration: "the disclosure of transcendence in any form of revelatory giving suggests that the mind submits in the end to imaging the unimaginable rather than remaining speechless in apophantic unknowing and aporetic suspension" (Wolfson 2014, p. 142). In the context of Levinas, Wolfson describes this dynamics as the "effacement of the nonphenomenolizable", which presents "too much of a hazard of making the anti-idolatry of formlessness into a form of idolatry" (ibid., p. 138).

23  On the difficult question of gender roles in Levinas, see (Chanter 2001; Elise Katz 2005, vol. III, pp. 190–211).

24  See Mekhilta of Rabbi Ishmael, Bahodesh, 8 (on Exodus 20, p. 14) for a midrashic reflection on the relation between the first and the sixth commandments: "I am the Lord your G-d," and opposite it "You shall not kill".

25  Hence the notion that my responsibility is infinite in the sense that it is "increasing in the measure that it is assumed; duties become greater in the measure that they are accomplished. The better I accomplish my duty the fewer rights I have; the more I am just the more guilty I am." (TI 274) This results from the fact that responsibility—my self-limitation vis-à-vis the Other—constitutes my own individual being.

26  Levinas (1988, p. 155) and Smith (1994, p. 133).

27  It is here that Levinas's logocentrism appears most clearly, to position him in direct tension with Derrida's *Grammatology*.

28  In the context of encounter between philosophers and prophets, more specifically in view of the history of the prophetic tradition, it is remarkable how Levinas's discourse problematizes the notion of "work" through the double figure of writing and state, of

letter and law. These two constitutive human creations, paradigms of civilization, have famously played a central role in Christian critique of Jewish reception of Prophecy, Paul and Luther being two canonic instances

29  My interpretation here is very much in line with the one offered by Robert Bernasconi, "Levinas's Ethical Critique of Levinasian Ethics", in (Davidson and Perpich 2012, pp. 253–70). Bernasconi rights pointed out that scholarship "tended to ignore that whole of the fourth section of *Totality and Infinity* on eros and fecundity" (p. 256) and indicates, in contrast, as I do, the crucial role of this section in the book's architecture, namely as "the fulfillment of the ethical relation" (p. 260), with "a political meaning" (p. 263).

30  Levinas did use the category of "nation" in other places, also affirmatively, for instance when he wrote about the Jews as a "nation that is united by ideas", see (Levinas 1954, p. 384; Hand 1990, p. 257).

31  Emmanuel Levinas, "Messianic Texts", *Difficult Freedom*, p. 149. On the "vexed question of Israel", see (Critchley 2004, pp. 172–85). My reading challenges Howard Caygill's appraisal that "in *Totality and Infinity* the translation of the work of justice into a political project is accomplished in a way that makes any identification of it with a state such as the 'State of Israel' extremely difficult"; see (Caygill 2002, p. 124).

32  Some commentators already noticed this affinity, see for instance (Large 2015), who however argued that Levinas's idea here is "entirely opposite" to Heidegger's, since in "the ethical community … the future is not ours, but the others'" (p. 111). This assessment, I argue, fails to take into account that Levinas's ethical community is based on fecundity, where the child is "at the same time other and myself".

33  See for example Brumlik (2018, pp. 41–54).

34  Arendt, *Origins*, p. 239.

35  Bernasconi, "Levinas's Ethical Critique of Levinasian Ethics", evokes the formal affinity of Levinas's "fecundity" to Heidegger's *Volk*, but characterizes Levinas's notion of "infinite of time" in opposition to Heidegger's "finitude of being" (p. 263). My argument undermines this opposition by showing how fecundity constitutes infinite time that is nonetheless individualized, i.e., genealogical, historical, and in this sense also finite. Bernasconi further interprets Levinas's notion of the "family" not as a paradigm of the nation, but of the private sphere, and for this reason not as contrasting but as agreeing with Arendt's "observation that totalitarianism destroys private life" (p. 265).

36  As Franz Rosenzweig showed, both performances of God's people generate the collective identity, the "We", as the agent of God's Kingdom.

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
