# Peer review of "Heidegger as Levinas’s Guide to Judaism beyond Philosophy"

_religions, doi:10.3390/rel12070477_

Round 1

Reviewer 1 Report

This is a strikingly original contribution to our understanding of Levinas's Totality and Infinity and for that reason I recommend it be published without hesitation. As is almost definitional, originality works against the grain of the standard understandings and, at least in this case, the article flies in the face of widespread reception of Levinas's thought. I am not convinced by its argument and I think the author is simply wrong on many points of interpretation. Three points in particular seem deeply wrong to me. One is the proposal that Levinas defends the individuality of the self on its own grounds, whereas most readers take him to grounding subjectivity on the claims of the Other. The second is the extensive use of the language of "epistemology" to explain Levinas's project, whereas most scholars read Levinas as defending an ethics that is pre-epistemic and inassimable to epistemic categories or language. The third is the account of the "family," which seems more Hegelian, and "the nation" which seems more Rosenzweigian, than Levinasian, since Levinas does not mention the family but views "election" as a filiation of singular but separated selves. 

I believe it is for the scholarly commuity to decide the merit of an interpretation and not one or two lone reviewers. The author presents his/her case clearly, sources the argument exegetically, and attempts to make his/her position cohere with other features of Levinas's work. Let readers of Religions decide its merits.  

Some points the author might consider before final submission.

1. The opening paragraph makes repeated reference to the “remorseful” reception of Heidegger. This strikes me as the wrong word. Levinas expresses no remorse for his debt to Heidegger, nor to other Jewish philosophers. They boldly admit it and honestly struggle with the implications of Heidegger's thought on their respective projects. 

2. I do not like the capitalization of "Philosophy," which neither Heidegger nor Levinas employ.

3. Re the discussion of Plato & Descartes: it seems to me that the erased figure here is Plotnus, though the point is extrinsic to this article.

4. Re the dicussion of “theory” – including fn 12. This goes back to Levinas's Theory of Intuition in Husser'l's Phenomenology and to Heidegger’s extensive critique of "theory" in SZ and related seminars, not least GA 27 which Levinas attended in person in 1928/29.

5. Lns 271-72 –  No doubt war destroys self and other, but the alternative to war begins, Levinas argues, with an admission of the legitimate claims of the other, and these ground and thereby "produce" the accountable self. The author seems to think that Levinas is defending the indivduality of the self on its own grounds. 

6. Ln 345 – I would have thought that the point of Levinas’s performative account of separation is precisely to resist inscribing the revelation of the other as a type of “knowledge”. I am not sure why the author insists on reading ethics as an “epistemology”. I would have thought the whole point is try to locate a normative force that does not rest on knowledge and is certainly not a theory of knowledge. Perhaps Cavell’s distinction between knowledge and acknowledgement would be helpful, but to read Levinas’s account of revelation as a type of knowledge seems to go against everything he is trying to do. In this respect Levinas is close to Kant of the second Critique. The force of morality does not rest on knowledge, for we can have no knowledge of free will. Kant too calls the force of practical morality “miraculous”. But the miracle is to be accepted, not known. At 365-70 the author says as much, but insists on the word “knowledge”. So too in speaking of an “epistemic tradition of the prophets”. Isn’t Levinas's point precisely to insist that prophecy is not epistemic but pre-epistemic, that the ‘here I am’ of the prophetic vocation answers to an “I know not what”? Why insist that this is knowledge? Better call it "faith", though this too is problematic. The phrase “a redeeming event of knowledge” seems to me to be positively opposed to Levinas’s thought. Interpreting Levinas’s thought as articulating an “epistemology of difference” strikes me as just wrong. Levinas’s point is that epistemology cannot found genuine difference, only a pre-epistemological separation produced through the non-knowledge of ethics can do that.

7. Lns 554-557, 559 are odd. Levinas develops an optics of the invisible, a voice of someone expressed at a distance that cannot be perceived in objective measures and is thus not objectively present but manifest only to the subjectivity.

8. Lns 758-65. This strikes me as completely antithetical to Levinas’s understanding of religion, which is not a first-person plurality (“we”) but a first-personal relation to a unique second-person, one that cuts across not only the state but also society. To read TI as a defense of “a people” is contrary to its stated intentions, according to which the defense of subjectivity is founded on a relation to the Other beyond the horizon of a community of people. Footnote 26 does not help. Fecundity is precisely not a relation to a people but to the singularity of the child/son.  Its political dimension is to transcend the horizon of collective existence in the name of a singular one who is still to come. I could not disagree more strongly with the interpretation of the family at lns 799ff. The family does not even appear as a concept in TI. The point of the erotic relation is that is constitutes an ambiguity of transcendence, ultimately a failure, for it falls back into an egoism en deux. To interpret Part IV of TI as a treatise on “the nation…unified in specific collective self-identity” is certainly novel, but I think most readers will agree that “election” has no collective significance for Levinas and refers, instead, to the sense of filiation, of being born as oneself under obligations that one has not oneself chosen. (If Levinas ever approaches the idea defended by the author it is under the sign of ‘destiny’ not ‘election’.)

Author Response

I thank and salute the reviewer for an exemplary show of academic integrity, and I appreciate that they do not hesitate to recommend publication despite their fundamental disagreement with the essay’s claims.

To the different points:

  1. I changed “remorseful” to “tormented”.
  2. I clarified why I write “Philosophy” in capital.
  3. Noted, thanks.
  4. Thanks for the references.
  5. Major point I: I do think that in many respects Levinas is worried about the effacement of the subject as a problem in itself – see the horror of the il y a in early texts, that are not yet predicated on ethics. In Totality and Infinity the argument is predicated on ethics, however the separate individual existence is a precondition for ethics, and this existence must be separate namely not existentially dependant on the Other, so that it may at all encounter the Other as Other. This is why the first part of the phenomenological narrative deals with “interiority”.
  6. Major point II: Levinas indeed criticizes the knowledge at the basis of Western philosophy, however his project does not seek to abandon knowledge, he is a philosopher and a talmudist. Rather he seeks to reconfigure knowledge. Especially in Totality and Infinity, ethics is not opposed to knowledge, on the contrary Levinas posits ethics as grounding knowledge: ethics is “optics”, ethics is “first philosophy”, revelation as “teaching”. What Levinas opposes is not any kind of knowledge, but theoretical knowledge, which is based on light and logos. This indeed goes back to his critique of Husserl’s exclusively theoretical or doxical paradigm of intentionality. So my point is that Levinas does not seek to offer, against the epistemology of Western philosophy, simply no episteme at all, but a different kind of knowledge and epistemology, namely ethics-based, which I claim he identifies in the Jewish tradition of knowledge, coded as “prophecy” in Totality and Infinity.
  7. I’m not sure what statements exactly are “odd”. However, the descriptions in Totality and Infinity do not suggest an encounter with invisibility, but with a “face”, which is not absent, but very much present, in fact constitutes presence, and not only “subjective”, but precisely objective, real – being and not phenomenon. That’s the whole point of ethics as “optics”. Levinas does not counter theoretical visibility with invisibility, but with another mode of vision, another optics and intentionality, namely the ethical. Ethics is a mode – the preliminary mode – of seeing, namely of perceiving an objective presence, being – as a face.
  8. I explicitly state that I offer a different reading than the common one articulated by the reviewer, and provide sufficient arguments and the references for my reading. I do not think religion “cuts across” society, but that the whole point of Section IV is to account for society “beyond the face”, i.e. beyond the merely individual encounter. The reviewer writes that “The family does not even appear as a concept in TI”. It does, and I quote it in my text, for instance: “The family does not only result from a rational arrangement of animality; it does not simply mark a step toward the anonymous universality of the State. It identifies itself outside of the State, even if the State reserves a framework for it.” (TI 342, 306). I do not agree with the claim that “election” has “no collective significance” for Levinas. His notion of “election” clearly echoes the Jewish notion, which is collective, and Levinas also refers to this collective notion in his Jewish writings, with respect to “Israel”.

Reviewer 2 Report

This is an excellently written and argued article.

If I were to make any suggestions for expansion, then perhaps I would suggest that the author's suggestion that the practical correlate for politics as Levinas imagines it would be Hannah Arendt's concept of "totalitarianism," is perhaps somewhat under-theorized. Either the author should omit this suggestion entirely, or should expand their reading of Arendt with more explicit citations of her work.

In addition, the writer gestures toward a messianic vision of "universalist particularism" at the end of the essay, without a clear definition of this term or explication of how this functions in Levinas's oeuvre. As Levinas borrowed this idea fairly explicitly from Rosenzweig, a brief detour into Rosenzweig's "Star of Redemption" would help orient the reader here.

Nonetheless, this is an excellent piece of work.

Author Response

I thank the reviewer for the reading, the kind evaluation and the good suggestions. It would definitely be important to expand the discussion on Levinas and Arendt, as well as on Levinas’s messianic politics of Israel. Nonetheless, in the framework of this essay, my feeling is that these discussions are not crucial for my central argument, so, since the text is already quite long, I decided to postpone these discussions for another, more suitable occasion.

Reviewer 3 Report

Overall summary:

-This essay examines Levinas’s Totality and Infinity in detail, drawing out how Levinas positions and develops Judaism (or the prophetic episteme) as the “other” of Westernized philosophy, and emphasizing how Heidegger inspires this conjunction or friendship as a dialogical opposition to social propensity toward totalization.

Broad Comments:

-Strengths: There is a close reading of necessary nuances for interpreting Levinas well, particularly the prophetic vs philosophic epistemes. This could be enhanced by glancing at David Klemm’s (1989) Levinas’ phenomenology of the other and language as the other of phenomenology, in Man and World, 22: 403–426. It breaks down Levinas’s performance in his own writing itself of prophetic and philosophic voices (Klemm identifies 4 voices rather than just 2). This is relevant for page 5, but also line 386 (“to perform an intervention”) and especially lines 456-468.

-Areas for improvement/weaknesses with suggestions:

  1. a) At times, the connection to Heidegger is lost—basically by page 7 or so, as a reader I forgot that this had to do with Heidegger inspiring Levinas, and the article became instead a great resource for Levinas exegesis. So I am not sure whether to suggest to reduce the Heidegger connection in the abstract and intro or to reduce the exegesis of TI and bring it back to Heidegger more explicitly throughout the rest of the paper.
  2. b) The writing style at times comes off as awkward or unfriendly to readers (see the sample suggested edits below).
  3. c) This may be from Levinas rather than the author – there seems to be an overemphasis on what is and is not philosophical. Line 102, for example, says the Talmud is not philosophical. It certainly contains philosophical discussions. Perhaps the kind of Western philosophy Heidegger and Levinas caricature is monologue, whereas the Talmud is full of commentarial dialogue?
  4. d) It may be useful to note Levinas’s myopic view of what counts as otherness, such as around lines 183-200 with the discussion of the “orient.” In Difficult Freedom, for example, Levinas writes: “Surely the rise of the countless masses of Asiatic and underdeveloped peoples threatens this new-found authenticity [of Jewish universalism]? On to the world stage come peoples and civilizations who no longer refer to our Sacred History, for whom Abraham, Isaac, and Jacob no longer mean anything . . . Under the greedy eyes of these countless hordes . . . the Jews and Christians are pushed to the margins of history, and soon no one will bother any more to differentiate between [them].” Or as Mortley reports Levinas saying “I often say, though it’s a dangerous thing to say publicly, that humanity consists of the Bible and the Greeks. All the rest can be translated: all the rest—all the exotic—is dance.” In Mortley, French Philosophers in Conversation (New York: Routledge, 1991), 18.

While these are later statements, they seem to culminate, not change, his thoughts about the orient discussed in TI.

Specific comments/edits:

-Line 36: I suggest adding “an” before “alternative”

-Line 47: I suggest adding “an” before “alternative” there… same as line 36. While some uses of “alternative” read smoothly, these two uses read oddly, and so as a reader I pause and then lose the point being made. I think it may be best to be readerly-friendly here rather than not.

-Line 74: Starting the paragraph with “in the quoted passage” makes readers like myself feel like… wait did I miss a long quoted passage? I think just leave that off and start with, “Levinas continues to qualify this acknowledgement…”

-Line 84: the “in the later Heidegger…” needs rewording to preserve the parallel, such as “which in the later Heidegger would…”

-Line 88: either say East Asia or China and Japan.

-Line 92: the wording is awkward—already saying Paul is a Saint, then adding, “also known for Christians, as the Bible.” Either just go straight for the Bible or stop at saying Paul’s letters. These awkward phrasings make the cadence seem like a profound point is being made where a reader should pay close attention.

-I’ll stop tracking these here but see throughout the rest of the paper similar writing issues.

Author Response

I thank the reviewer for the reading, the kind evaluation and the good suggestions. To the specific points:

Thanks for the reference to Klemm’s text, which I was not aware of. I added a reference.

To 1. It is true that the essay can also work as an independent reading of Totality and Infinity. However, my original reading of Levinas (the inter-epistemic drama, thought beyond philosophy etc.) is easier to understand within the context of the Heideggerian project, which I claim critically inspired Levinas. This is why I think it still makes sense to frame my reading with the influence of Heidegger.

To 2. Thanks for the suggested stylistic changes, which I adopted. I further re-read the text and made more changes to make reading easier.

To 3. It is indeed my claim that Levinas’s overall project constructs an epistemic difference between the tradition of Western Philosophy and the tradition of Jewish Talmud. This difference does not mean that they are completely foreign to each other. On the contrary, Levinas thinks both traditions try to do something similar, and for this reason Talmud is not anti-philosophical, but ultimately helps philosophy better attain its own goal, epistemically and ethically.

To 4. Thanks for the good point about the “orientalists” future of Levinas’s thought. I added a note on this.